# Area-Level Walkability and the Geographic Distribution of High Body Mass in Sydney, Australia: A Spatial Analysis Using the 45 and Up Study

**DOI:** 10.3390/ijerph16040664

**Published:** 2019-02-24

**Authors:** Darren J. Mayne, Geoffrey G. Morgan, Bin B. Jalaludin, Adrian E. Bauman

**Affiliations:** 1The University of Sydney, School of Public Health, Sydney, NSW 2006, Australia; geoffrey.morgan@sydney.edu.au (G.G.M.); adrian.bauman@sydney.edu.au (A.E.B.); 2Illawarra Shoalhaven Local Health District, Public Health Unit, Warrawong, NSW 2502, Australia; 3University of Wollongong, School of Medicine, Wollongong, NSW 2522, Australia; 4Illawarra Health and Medical Research Institute, University of Wollongong, Wollongong, NSW 2522, Australia; 5The University of Sydney, University Centre for Rural Health, Rural Clinical School—Northern Rivers, Sydney, NSW 2006, Australia; 6Ingham Institute, University of New South Wales, Sydney, NSW 2052, Australia; b.jalaludin@unsw.edu.au; 7Epidemiology, Healthy People and Places Unit, Population Health, South Western Sydney Local Health District, Liverpool, NSW 1871, Australia

**Keywords:** body mass, disease mapping, geographic variation, obese, overweight, spatial analysis, walkability

## Abstract

Improving the walkability of built environments to promote healthy lifestyles and reduce high body mass is increasingly considered in regional development plans. Walkability indexes have the potential to inform, benchmark and monitor these plans if they are associated with variation in body mass outcomes at spatial scales used for health and urban planning. We assessed relationships between area-level walkability and prevalence and geographic variation in overweight and obesity using an Australian population-based cohort comprising 92,157 Sydney respondents to the 45 and Up Study baseline survey between January 2006 and April 2009. Individual-level data on overweight and obesity were aggregated to 2006 Australian postal areas and analysed as a function of area-level Sydney Walkability Index quartiles using conditional auto regression spatial models adjusted for demographic, social, economic, health and socioeconomic factors. Both overweight and obesity were highly clustered with higher-than-expected prevalence concentrated in the urban sprawl region of western Sydney, and lower-than-expected prevalence in central and eastern Sydney. In fully adjusted spatial models, prevalence of overweight and obesity was 6% and 11% lower in medium-high versus low, and 10% and 15% lower in high versus low walkability postcodes, respectively. Postal area walkability explained approximately 20% and 9% of the excess spatial variation in overweight and obesity that remained after accounting for other individual- and area-level factors. These findings provide support for the potential of area-level walkability indexes to inform, benchmark and monitor regional plans aimed at targeted approaches to reducing population-levels of high body mass through environmental interventions. Future research should consider potential confounding due to neighbourhood self-selection on area-level walkability relations.

## 1. Introduction

The increasing prevalence of overweight and obesity is a universal and urgent public health problem [1]. High body mass index ≥25 kg/m^2^ (overweight or obese) contributed 5.7% of total disability adjusted life years (DALY) to the global burden of disease in 2016, making it the fifth leading risk factor—up from 2.7% of total DALYs and a ranking of 12 in 1990 [2]. High body mass is a risk factor for cardiovascular disease, cancer, type 2 diabetes mellitus, and musculoskeletal conditions [3,4], while its economic costs to health care systems and communities grow with increasing levels of overweight and obesity [5]. The physiological energy imbalance that underlies high body mass is influenced by genetic, behavioural, social, economic, and environmental factors operating within multiple complex systems [6,7,8]. Reducing the health and economic burdens of overweight and obesity will require shifts in these population-level systems [7]. For example, environmental interventions that typically produce small individual-level effects may aggregate into large population-level benefits because exposure is ubiquitous [8,9] and relatively persistent [10,11].

The built environment refers to that “part of the physical environment...Constructed by human activity” ( [12] p. S550), and is hypothesised to contribute to high body mass by influencing lifestyle behaviours that underlie its development [8]. The emerging consensus from the extensive literature is that the built environment evidence base is sufficiently developed to incorporate into planning, policy and interventions aimed at reducing high body mass [7,13], although uncertainties remain (see reviews by [14,15,16,17,18]). ”Walkability” describes the capacity of the built environment to promote walking for multiple purposes [19], and may contribute to reducing overweight and obesity by promoting participation in total daily moderate-intensity physical activity [8,9,17,20,21,22]. To this end, it is increasingly considered in development plans aimed at enhancing physical and social infrastructure to promote healthy lifestyles and reduce the burden of chronic conditions like high body mass on populations (e.g., [23,24,25,26]).

Walkability indexes have been identified as potentially useful tools for planning, benchmarking and monitoring environmental policies and interventions to improve walkability, and translating the outcomes of walkability research from rhetoric to action [27,28]. While numerous indexes exist (e.g., [29,30,31,32,33,34,35,36,37]), the Neighborhood Quality of Life Study (NQLS) [38] and Physical Activity in Localities and Community Environments (PLACE) Study [19] indexes remain the most influential [39]; underpin a majority of research linking walkability to health behaviours and outcomes, including high body mass [17]; and are applicable in planning, policy and practice settings [28], which is facilitated by their capacity to be constructed at multiple spatial resolutions [38,40]. These indexes operationalise walkability using residential density, street connectivity, land use mix and, if available, retail floor area ratio, destinations or density within a geographic information system [19,38,40]. Index variables serve as proxy measures for built environment attributes associated with walking. Land use mix measures the diversity and concentration of land uses in an area, while intersection density measures the directness of paths [38]. Compact areas with diverse land uses that are highly connected promote walking by reducing the distance between origins and destinations [9,19,40]. Similarly, high population densities provide critical masses that concentrate diverse destinations within compact areas [12,38,40], and is measured by residential density. The ratio of retail floor area to retail land use is a measure of pedestrian access, with larger values indicating greater area given to pedestrian uses and less area to car parking [41].

The extent to which walkability indexes are associated with health outcomes at population scales is a key consideration in their utility to benchmark and guide planning and policy aimed at reducing population-levels of overweight and obesity [27,42]. This is because health and urban planning that influences environmental walkability occurs at local, urban and regional scales [43,44]; that is, for communities and populations. These meso (neighbourhoods/communities) and macro (cities) geographic scales are much coarser than typically used in studies to derive built environment exposure-response evidence [42,45], which mostly focus on individual (micro) level risk (see reviews by [14,15,16,17,22,46]). Measured at the micro-scale, walkability is typically derived within a radial or street-based network buffer of 200–1600 metres around a residential address; reflects the immediate built environment to which an individual is exposed [47,48]; and is preferred for individual-level research.

In contrast, when measured at meso- and macro-scales, walkability (or sprawl) is usually calculated within an administrative boundary; represents a contextual variable describing the shared built environment to which groups, communities and populations are exposed; and is especially useful for area-level (ecological) research and planning applications [42]. Using individual-level walkability evidence to inform activity at coarser planning scales has raised concern in the literature for its potential to result in flawed public health action [49]. This is a concern about atomistic [50] or individualistic [51] fallacy, which is the area-level corollary of the ecological fallacy and refers to the erroneous use of data on individuals to make inferences about groups [52].

Analysis at the geographic scale of planning addresses concern about erroneous cross-level inference [42], and has been identified as highly relevant to “local area” walkability planning because it produces evidence at the level where decisions are made [43]. Rydin and colleagues have also identified the need for “urban scale” data to inform planning and policy interventions that maintain the urban advantage in health outcomes [44]. However, studies that match walkability exposures and body mass outcomes at these planning scales are uncommon [16,17,46], despite calls from planners and policy makers for evidence at this level [45,53,54,55]. What evidence is available at these planning scales comes largely from ecological analyses in the United States, which have consistently found higher body mass and prevalence of obesity in sprawling versus compact counties (e.g., [30,53,56,57,58]). Compactness at this scale is generally considered synonymous with more walkable environments. However, sprawl indexes have been criticized for conflating multiple built environment concepts, and not providing a coherent, unitary measure of walkability [38,40].

Geographic variation in overweight and obesity has been reported within many countries [59], and needs to be considered in health programming and planning. In addition to identifying areas at increased risk of adverse health outcomes [59,60], geographic variation in excess of that due to known factors can indicate place-based influences on health that are distinct to the contextual effects arising from differences in the demographic, social and economic composition of populations between areas [61,62]. Spatial analysis is particularly useful in identifying these place-based effects because it quantifies the contribution of both observed and unobserved factors to geographic variation in outcomes while accounting for spatial autocorrelation that can lead to biased statistical inference [62]. In this context, geographic variation encompasses more than just differences between areas, which is well addressed in the body mass literature (e.g., [63,64,65,66,67,68,69,70,71,72]). It is also the spatial expression or distribution of this variation [73]. Spatial analysis is concerned with location [74]. It leverages the underlying process giving rise to the geographic variation rather than reducing it to a naïve dummy-coded comparison of areas in the case of fixed-effects analysis, or focusing on reductions in intraclass correlation coefficients that conflate spatial and non-spatial sources of variation through a common random effect term as in multilevel analysis [42].

Spatial analysis has the potential to provide unique information on relations between walkability and high body mass. For example, we have previously reported that physical activity is geographically structured in Sydney, and that area-level walkability accounts for some of this spatial patterning [42]. The ”disease mapping” approach [75] used in this study also produces smoothed maps that can be used to communicate spatially varying risks to planners, policy-makers and other interested stakeholders [42]. Identifying spatial disparities in contextual factors that contribute to adverse health outcomes at appropriate intervention scales has been identified as essential for informing place-based interventions aimed at improving population health [76]. Spatial analysis is uniquely placed to assist in addressing these disparities and environmental inequalities through its capacity to identify and target geographic areas where environment-related health risks are disproportionately higher and potentially amenable to intervention [42,77]. However, despite an increasing use of geographic information systems in the high body mass literature, the application of spatial analysis at any scale is rare [78]. For example, only a few area-level studies have used an explicitly spatial approach to address geographic variation in overweight and obesity [56,59,62,72,79,80,81,82,83,84,85,86,87,88,89]; an even smaller number have considered built environment influences on this geographic variation [62,81,82,89]; and only one appears to have evaluated the contribution of walkability to this geographic variation directly [81].

The objective of this study was to build on our previous work in the Sydney statistical district [39,42] and assess relations between area-level walkability and population-levels of overweight and obesity using an explicitly spatial approach, and at a geographic scale representative of those used for “local area” [43] and “urban scale” [44] planning. The specific aims of the study were to (i) assess area-level associations between walkability and prevalence of overweight and obesity in Sydney; (ii) assess geographic variation in area-level prevalence of overweight and obesity in Sydney; and (iii) assess the extent to which area-level walkability accounts for geographic variation in overweight and obesity in Sydney beyond that due to individual-level demographic, social, economic and health factors, and area-level socioeconomic disadvantage.

## 2. Materials and Methods

### 2.1. Study Design and Area

We investigated associations between area-level walkability and prevalence of overweight and obesity in the Sydney statistical division of New South Wales, Australia [90], using a cross-sectional ecological study design, which is appropriate and valid for area-level inference [52]. Sydney has a land area of 12,142 km^2^, and was Australia’s most populous city at the 2006 Australian Census with an estimated resident population of 4.1 million people living in 1.6 million dwellings [91]. We used Census postal areas as our units of analysis to coincide with the smallest spatial unit at which individual-level data were geographically identified by the data custodian. In 2006 there were 260 conterminous postal areas across the Sydney statistical division [92] with median and inter quartile range (IQR) values for land area of 7.6 (IQR = 3.7–19.4) km^2^, 5304 (IQR = 2694–8426) residential dwellings, and 13,090 (IQR = 6529–22,092) residents [91]. The median land area of postal areas corresponds to a radial buffer of 1550 m, which is within the range of buffer sizes for which consistent environment-behaviour associations have been reported in individual-level studies of walkability [47,48], and is likely a reasonable analogue of the “local areas” and “urban scales” at which health and urban planning decisions occur [43,44].

### 2.2. Participants

Individual-level data used in this study were obtained from participants of The Sax Institute 45 and Up Study [93] approved and monitored by the University of New South Wales Human Research Ethics Committee (ref no. HREC 05035/HREC 10186). This population-based cohort study was established to investigate healthy ageing in the New South Wales population aged 45 years and over [93]. Study recruitment occurred between January 2006 and December 2009 [94] for a final cohort size of 267,153 participants or approximately 10% of the total New South Wales target population [95]. Eligible persons were randomly sampled from the the Department of Human Services (formerly Medicare Australia) enrolment database. Selected individuals were mailed an invitation letter, and asked to return a signed, written consent form with their baseline survey via reply-paid mail if they consented to participating in the study [93]. We were provided access to the April 2009 data release, which the data custodian had geocoded to 2006 Census statistical divisions and postal areas. We limited our analysis to 115,153 respondents living in the Sydney statistical division to coincide with the spatial extent of our exposure variable. Our research comprised a sub-study of the Social, Environmental, and Economic Factors Study approved and monitored by the University of Sydney Human Research Ethics Committee (ref No. 10-2009/12187). Details on accessing 45 and Up Study data are available on The Sax Institute website (www.saxinstitute.org.au/our-work/45-up-study).

### 2.3. Data

Individual-level data included self-reported responses to the baseline survey of 45 and Up Study collected between January 2006 and April 2009 [93], which we used to calculate and adjust area-level outcome variables. Postal area contextual variables comprised the Sydney Walkability Index (SWI) [40] and 2006 Index of Relative Socioeconomic Disadvantage [96], which we included as study and covariate factors, respectively.

### 2.4. Outcome Variable

The primary outcome measures used in our study were self-reported overweight and obesity, which we defined using the standard body mass index (BMI) formula of weight in kilograms (kg) over height in metres (m) squared (kg/m^2^) and World Health Organisation (WHO) cut-points of 25.0–<30.0 kg/m^2^ for overweight and ≥30.0 kg/m^2^ for obesity [97]. Self-reported BMI has been validated against measured BMI as a generally appropriate method for quantifying body size in the 45 and Up Study cohort, although it is known to underestimate prevalence of obesity when classified using standard BMI categories [98]. Overweight and obesity status were represented as dichotomous (yes/no) variables for individual-level analyses, and as counts of overweight and obese respondents within postcodes in area-level analyses.

### 2.5. Exposure Variable

The exposure variable used for all analyses was postal area walkability, which we measured using the Sydney Walkability Index [40]. This three-factor index is derived using methods and data comparable to the PLACE and NQLS walkability index [19,38]. The Sydney Walkability Index is calculated within a geographical information system using three built environment variables: residential dwelling density (the number of residential dwellings per square kilometre of residential land use); intersection density (the number of intersections with three or more roads per square kilometre of total land area); and land use mix (the entropy of residential, commercial, industrial, recreational and other land uses). The Sydney Walkability Index was derived at the 2006 postal area level using 2007 spatial data to temporally align it with the midpoint of the of the 45 and Up Study baseline data collection.

Environmental variable values are divided into deciles, scored from 1 (lowest) to 10 (highest), summed to give a total score between 3–30, and then divided into quartiles corresponding to low, low-medium, medium-high and high walkability [40]. Environmental values increase monotonically within strata and have median values of 2.3, 13.4, 19.8 and 46 dwellings per hectare for residential density; 3.4, 46.1, 79.5 and 162.5 intersections per square kilometre for street network connectivity; and entropies of 0.005, 0.033, 0.056, and 0.134 for land use mix (see [42]). The Sydney Walkability Index has predictive validity for utilitarian walking, is comparable to four-variable indexes in the research literature, and is associated with population-levels of moderate and vigorous physical activity [40,42].

Walkability was entered as an index in our analysis for consistency with the interest expressed in the literature on using “walkability indexes” to benchmark, inform and monitor regional development plans [27,28], and because the non-parametric functions used in other studies [99,100,101] to model index components separately would have made our already computationally-intensive spatial analyses intractable.

### 2.6. Covariates

Individual- and area-level factors from the 45 and Up Study and substantive literature likely to contribute to, or confound, associations between walkability and body mass were included as covariates in our analysis (see [102,103,104,105,106,107,108,109,110,111,112,113]). Individual-level covariates included self-reported sex; five-year age group at baseline interview; language spoken at home; educational level; relationship status; employment status; health insurance type; level of psychosocial distress measured using the Kessler Psychological Distress Scale [114] (minor, moderate, high, very high [115,116,117]); smoking status; number of chronic conditions ever diagnosed and treated in the previous four weeks; and functional capacity, which was measured using the Medical Outcomes Study (MOS) 36-Item Short-Form Health Survey (SF-36) physical functioning scale [118,119] and classified as none (0 to <60), minor (60 to <90), moderate (90 to <100), and severe (100) [120]. Postal area socioeconomic disadvantage was measured using the Index of Relative Socio-economic Disadvantage from the 2006 Australian Census [96]. We did not include physical activity in our analysis because it likely mediates relations between the built environment and high body mass [8,14,15,16,17].

### 2.7. Statistical Analysis

The objective of our analysis was to assess relations between walkability and the prevalence and geographic distribution of overweight and obesity in the Sydney statistical district at a scale analogous to those at which health and urban planning decisions are made. This objective is appropriately addressed by an ecological (spatial) analysis because the targets of inference are areas, not individuals [52]. We have previously identified high levels of spatial autocorrelation in 45 and Up Study data that have both research and planning utility, and the potential to bias inference if not addressed in the analysis [42]. Multilevel models can account for spatial autocorrelation but typically conflate spatial and non-spatial variation through a common variance component [42]. We therefore explicitly modelled the underlying spatial and non-spatial sources of variation in our data using a relative risk implementation of the ecological Besag, York and Mollié (BYM) conditional auto regressive model.

The BYM is a fully Bayesian ecological spatial model fit to aggregate data, which is commonly used in epidemiology for “disease mapping” applications [75]. The goal of disease mapping is to recover a map displaying variation in the geographic distribution of risks for spatial units within a study area that is “smoothed” of extreme and unreliable estimates that can arise from differences between units in the sizes of their underlying populations [75]. This is achieved in the BYM model by decomposing map variation into an unstructured variance component that smooths risk estimates towards the global mean of the study area, and a spatially structured (geographic) variance component that smooths risk estimates towards the local mean of contiguous spatial units [75,121]. These components also indicate the extent to which map variation is due to structured (spatial) and unstructured (non-spatial) factors.

The BYM model can be extended to ecological regression problems by incorporating area-level covariates into its specification [75], but it cannot parsimoniously control for individual-level factors that may confound area-level effect estimates. We therefore used a two stage modelling strategy adopted by other researchers in the epidemiological literature whereby individual-level regression models are used to estimate expected cases for each outcome, which are then used as offset terms in area-level spatial analyses to adjust for the varying size and composition of populations between spatial units (see [39,42,122,123,124]).

In the first step, we estimated the predicted log odds ($l_{ij}$) of overweight and obesity for individuals using conditional fixed-effects logistic regression models:(1)$${\hat{l}}_{ij}=\alpha +x_{i}\beta$$
where ${\hat{l}}_{ij}$ is the predicted log odds of being either overweight or obese for the $i^{th}$ person in the $j^{th}$ postal area, α is the model intercept, and $x_{i}\beta$ is an optional vector of individual-level covariates. We fit two models for each outcome: (1) an unadjusted null model with no covariates; and (2) an adjusted model including all individual-level covariates described previously. The log odds for individuals from each model were converted to a predicted probability using the inverse link function:(2)$${\hat{Y}}_{ij}=\frac{e^{l_{ij}}}{1+e^{l_{ij}}}$$
We then summed these probabilities within each postal area to obtain the expected number of cases for each outcome based on (1) the prevalence in the study area from unadjusted logistic regression estimates; and (2) the underlying respondent structure of our sample from adjusted logistic regression estimates (see [39,42,122,123,124]). These expected case counts were used as offsets in the spatial Poisson regressions described in step 2, and are referred to as unadjusted and adjusted offsets, respectively.

In the second step, we used relative risk implementations of the BYM model with Poisson likelihoods to estimate prevalence ratios for postal areas relative to the study area [125]. The BYM model is a fully Bayesian spatial model fit to aggregate data that decomposes total variation into observed and unobserved sources [75,121] using:(3)$$log\left(\theta_{j}\right)=\alpha +x_{j}\beta +s_{j}+u_{j}+log\left(e_{j}\right)$$
where $\theta_{j}$ is the prevalence ratio for the $j^{th}$ postal area; α is the prevalence ratio for the study area; $x_{j}$ and β are vectors of observed area-level explanatory variables and associated regression parameters estimates; $s_{j}$ and $u_{j}$ are unobserved spatially structured and unstructured random effects; and $e_{j}$ is an offset term representing the expected number cases in the $j^{th}$ area. The unstructured variance ($u_{j}$) is a normal independent and identically distributed residual, while the spatial variance ($s_{j}$) is conditionally normally distributed on the mean prevalence of the surrounding *k* contiguous postal areas [75]. Model offsets ($e_{j}$) corresponded to those derived for postal areas in step one, and were either unadjusted or adjusted for individual-level factors.

The total count of overweight and obese respondents ($o_{j}$) in each postal area served as the dependent variable in each model. We fit six BYM spatial regressions for each outcome: (1) a null model with unadjusted offsets; (2) a null model with adjusted offsets; (3) a covariate model with adjusted offsets and postal area walkability; (4) a covariate model with adjusted offsets and postal area socioeconomic disadvantage; (5) a covariate model with adjusted offsets and postal area walkability and socioeconomic disadvantage, and (6) an effect modification model with adjusted offsets, postal area walkability and socioeconomic disadvantage, and their interaction. A total of 10,000 draws from the posterior distributions of two Monte Carlo Markov Chains sampled every 250th iteration were used to obtain medians and 95% credible intervals for each model. Chain convergence was assessed using autocorrelation plots and the Gelman-Rubin diagnostic [126]. We chose between alternate models using the *Deviance Information Criterion* (DIC) [127], and mapped exponentiated linear predictors and variance estimates using quintiles to visualise geographic variation in risk of high body mass. The spatial fraction (ρ) for each model was calculated from the marginal variances of the random effects, and used to index the proportion of residual variation due to unobserved spatial factors (i.e., $\sigma_{s}^{2}/[\sigma_{s}^{2}+\sigma_{u}^{2}]$) (see [128,129]). Models were fit in WinBUGS 1.4.3 using R 3.3.2 and unweighted survey data, which produce unbiased, representative and generalisable relative effect estimates for individual- and area-level analyses in this cohort [42,130,131].

## 3. Results

Complete data were available for 92,157 of 115,153 (80.0%) Sydney respondents residing in 254 of 260 (97.7%) study postal areas. The median number of respondents per postal area was 212, and ranged from 0 to 2532 with an inter-quartile range of 110–363. Individual-level attributes for respondents included in analyses are shown in the *Characteristics* section of Table 1. Consistent with the larger 45 and Up Study cohort [132], our sample had a similar gender and employment profile to the study area but was otherwise younger, more highly educated, less likely to speak a language other than English at home, and more likely to be living with a partner than the general Sydney population aged 45 years and over [91].

### 3.1. Prevalence Overweight and Obesity

The within-cohort prevalence of overweight and obesity were 49.0% (48.7–49.4%) and 33.6% (33.2–34.0%), respectively. Table 1 reports prevalence by area- and individual-level characteristics. Prevalence of both overweight and obesity were highest in postal areas with low walkability, lowest in postal areas with high walkability, and displayed a exposure-response gradient. Likewise, overweight and obesity reduced with reducing levels of postal area socioeconomic disadvantage. For individual-level factors, overweight and obesity were more prevalent in males, persons speaking English at home or living with a partner, less educated individuals and full-time workers, persons without private health insurance, and past smokers; and increased with age to 65–69 years, psychosocial distress, number of diagnosed and treated chronic health conditions, and reduced functional capacity.

### 3.2. Individual-Level Factors

Table 2 shows adjusted fixed-effects estimates for overweight and obesity used to derive adjusted offsets for postal area spatial models. All effects were significantly associated with body mass outcomes and mostly consistent with the prevalence patterns reported in Table 1. The stand-out exception was a reversal in gradient between obesity and psychosocial distress from positive to negative after adjustment. This was due to confounding by functional capacity, which was both an independent risk factor for obesity (see Table 2) and strongly associated with psychosocial distress ($\chi {}_{9}^{2}$ = 4072.4, *p* < 0.0001). Other notable differences following adjustment were relationship status, which was unrelated to either overweight or obesity; age, which became associated with monotonically decreasing odds of obesity across the lifespan; and smoking status, which became associated with reduced odds of obesity for current compared to non smokers.

### 3.3. Spatial Analysis

Table 3 and Table 4 report parameter estimates and diagnostics for spatial regressions fit to overweight and obesity data. Smoothed prevalance ratios for postal areas from unadjusted null models ranged from 0.83–1.16 for overweight and 0.46–1.68 for obesity. Variation in risks between postal areas was principally due to unobserved spatial factors, with >96% of residual variation attributed to the spatial variance component for both overweigtht and obesity (see spatial fractions for Model 1 in Table 3 and Table 4). Adjusting for individual-level factors (Model 2) attenuated the ranges of smoothed prevalence ratios to 0.88–1.08 for overweight and 0.63–1.23 for obesity, but had little effect on the proportions of residual variation from spatial sources, which remained high at >93% for both outcomes. Univariable parameter estimates for area-level associations including walkability and socioeconomic disadvantage are shown in the Model 3 and 4 columns of Table 3 and Table 4. Risk ratios for walkability indicated consistent exposure gradients for both outcomes, with prevalence of overweight reduced by 4% and 9% and obesity by 8% and 11% in medium-high and high versus low walkability postal areas. Likewise, high body mass reduced monotonically with decreasing socioeconomic disadvantage. Overweight was 6% lower in the least versus most disadvantaged postal areas, and obesity was 11% and 9% lower in the least and second-to-least versus most disadvantaged postal areas. Fully-adjusted spatial regressions including individual- and area-level factors (Model 5) had the lowest DIC values and were the best fitting models for both outcomes (see DIC row in Table 3 and Table 4). Prevalence ratios for socioeconomic disadvantage in these models were largely unaffected; however, gradients for area-level walkability strengthened with overweight 6% and 10% lower and obesity 11% and 15% lower in medium-high and high versus low walkability postcodes. These fully-adjusted spatial models also had the smallest amounts of residual spatial variation, with 67% of unexplained model variation attributed to unobserved spatial factors for overweight and 90% for obesity. Interaction analyses (Models 6) provided no evidence that the observed associations between walkability and overweight ($DIC_{M6}-DIC_{M5}=$ 18.21) or obesity ($DIC_{M6}-DIC_{M5}=$ 12.12) were modified by postal area socioeconomic disadvantage.

### 3.4. Prevalence Maps

Figure 1 and Figure 2 graphically display smoothed prevalence ratios for overweight and obesity obtained from spatial models 1 (unadjusted null model), 2 (adjusted null model) and 5 (adjusted model with walkability and socioeconomic disadvantage). Total excess prevalence is shown in Maps A, D and G, and decomposed into excess risk due to spatial factors in maps B, E and H, and unstructured factors in maps C, F and I. Two features stand-out in each set of maps. First, residual prevalence is principally due to unobserved place-based factors, with higher ratios in spatial (B, E and H) versus unstructured (C, F and I) maps; and second, this geographic variation in risk reduces as individual- (Model 2) and area-level (Model 5) factors are added to spatial models. In unadjusted (Model 1) and adjusted (Model 2) null models, higher-than-expected prevalence was concentrated in western Sydney, and lower-than-expected prevalence in central and eastern Sydney. Including area-level walkability and relative socioeconomic disadvantage (Model 5) substantially attenuated excess prevalence by reducing excess risk attributable to unobserved spatial factors (see maps G and H of Figure 1 and Figure 2). Final excess prevalence estimates were reduced in western Sydney and the peri-urban fringe for both overweight and obesity; and remained higher-than-expected for obesity through south-central Sydney, and lower-than-expected for both outcomes on the eastern seaboard north of the Sydney central business district.

## 4. Discussion

This is one of only a small number of studies to examine geographic variation in high body mass and its association with environmental walkability using a large population-derived cohort and spatial analytic framework. We find strong support for associations between postal area walkability and area-levels of overweight and obesity among persons aged 45 years and over living in Sydney, Australia. Prevalence in postal areas with medium-high and high walkability is reduced by 6% and 10% for overweight and 11% and 15% for obesity compared to postal areas with low walkability, and are independent of individual-level social, economic and health status factors, and area-level socioeconomic disadvantage. We also find that both overweight and obesity are geographically clustered at the postal area level with lowest prevalence in and to the north of the central business district, and highest prevalence in western Sydney. Postal area walkability explains approximately 20% and 9% of this geographic variation in overweight and obesity, respectively, that is not attributable to individual-level factors and area-level socioeconomic disadvantage. Our findings confirm associations between high body mass and walkability at spatial scales typical of those used for public health planning; highlight the potential for spatial analysis to better integrate “place” into walkability research; and provide novel methods and data for New South Wales Government initiatives aimed at creating built environments that support active transportation and promote healthy lifestyles, and monitoring these initiatives.

Despite some limitations, the existing built environment evidence base appears sufficiently developed to inform interventions aimed at addressing high body mass at individual and population levels [13]. A recent review concluded that the strongest evidence is for meso- and macro-level correlates, and identifies urban sprawl, land use mix, street connectivity, population density, and proximity to services and destinations as the important environmental characteristics at these levels [17]. Walkability indexes combine many of these key environmental variables into summary metrics that can be easily implemented at multiple spatial scales for planning purposes [19,27,28,40]. Our study is novel because it directly addresses exposure-outcome relations at a geographic scale more proximal to those typically used by government agencies for population-level health and urban planning [43,44,45]. We observed that higher levels of postal area walkability measured by our index were associated with lower prevalence of overweight and obesity in postal area populations aged ≥45 years, even after adjusting for other individual- and area-level characteristics related to body mass. These findings coincide with the small but consistent body of area-level findings that increased body mass and prevalence of obesity are associated with greater urban sprawl (see review by [17]), and extend recent individual-level associations between walkability and body mass [133,134,135,136,137] to populations and the spatial scales at which health and urban planning decisions are made. Our study also provides new evidence on the potential of tools like the Sydney Walkability Index [40] to benchmark, inform and monitor health and urban planning activities aimed at reducing population-levels of overweight and obesity. This will have relevance in the Australian context where open-access tools have been developed that allow researchers and planners to calculate NQLS-PLACE index values at mutiple spatial scales (see [28,138]).

Action to address overweight and obesity should target populations of greatest need [59,60]. However, it is unlikely that at-risk groups will be uniformly distributed across an area such as the Sydney statistical district [88]. Spatial analysis is especially useful in this regard with its ability to identify geographic locations with higher (or lower) than expected rates of overweight and obesity, and whether this variation is explained by, or in addition to, factors known to influence the distribution of health, such as demographic and socioeconomic characteristics [61]. We observed very strong clustering of overweight and obesity through central Sydney that was due to unobserved and spatially structured factors, and which contributed the majority of excess risk. Including individual-level demographic, social, economic and health status factors in our analysis attenuated excess prevalence of high body mass and reduced spatial variance but had little effect on outcome clustering across the study region. This is consistent with Canadian findings that individual-level variables were important correlates of within-region variation but explained little between-area geographic variation [85]. In contrast, adding postal-area walkability and socioeconomic disadvantage to our models reduced area-level clustering of overweight and, to a lesser extent, obesity. However, our final maps remained weakly clustered. This residual variation could suggest the presence of other unobserved spatial factors structuring the residual prevalence of high body mass in our study area. Identifying these additional factors was beyond the aims of our study but may include greenspace, access to shops and services, aesthetics, the food environment, and proximity to public transport [7,8,13,62]. It is also possible that some of this residual variation is due to residual confounding of associations between walkability and high body mass by sociodemographic factors. For example, Frank and colleagues have reported that features of walkable neighbourhoods are associated with lower overweight in males but greater overweight in females without a degree, and with lower obesity in men with a degree but higher obesity in unemployed non-white men without a degree and white women without a degree [139]. Likewise, there is some evidence that higher body mass is negatively associated with features of walkable neighbourhoods in high socioeconomic communities, and positively associated in low socioeconomic minority communities [140]. Future spatial studies employing our approach should consider alternate adjustment techniques to account for this possibility; for example, by calculating offsets using a logistic machine learning classifier.

Our findings are consistent with a growing evidence base indicating geographic variation at multiple spatial scales in the distribution of overweight and obesity that have relevance for health, urban and transport planning [56,59,62,79,80,81,82,83,84,85,86,87,88,89], although only a few studies have investigated built environment correlates of this variation within a geospatial context [62,81,82]. While Shuurman and colleagues were unable to assess whether population density—included in most walkability indexes—patterned obesity in Metro Vancouver because obesity itself did not cluster in their study area [82], Congdon has reported that not only are obesity rates 13–20% higher in sprawling versus compact US counties—an effect size similar to that obtained for physical inactivity, but that adding environmental measures to spatial models of county-level obesity prevalence reduced unexplained spatial variation by 22%. Lathey et al. have also examined associations between obesity rates and sprawl factors, including walkability, for census blocks in Maricopa County, Arizona [81]. They defined walkability as accessibility to places of social interaction, and found it was the strongest model predictor of being in a ”high disease” obesity cluster with odds halved for the most versus least walkable census blocks [81]. Cluster membership was also associated with residential and commercial land use, and street connectivity, although effect sizes were very small [81]. Unfortunately, the focus on correlates of cluster membership reduces the analysis to a consideration of between-group differences, which is not especially informative geographically. Our study adds to the evidence base by its explicit focus on walkability and its contribution to geographical variation in high body mass at the spatial scales where health and urban planning decisions are made. We found effect sizes for walkability that were meaningful at population-scales [10], and sizable reductions in unexplained spatial variation comparable to other area-level spatial analyses [62].

Despite substantial reductions in unexplained variation due to spatially structured factors of 93.6% for overweight and 89.1% for obese, we observed little impact on spatial fractions except for overweight model 5, which reduced from ≥88.2% (models 1–4) to 67.3%. This is not surprising given the unstructured variance reduced by just 12.9% for overweight and 46.1% for obesity over the range of models fitted; with most of this decrease occurring between models 1 and 2 when we first adjusted for individual-level factors. Lunn et al. have noted that either the spatial (*s*) or unstructured (*u*) variance component will typically dominate the other in practical implementations of the Besag, York and Mollié model but will only be apparent once the posterior distributions of the components are examined [141]. A key strength of the Besag, York and Mollié model is its robustness to spatial and non spatial variation, and will produce unbiased parameter and variance estimates in the absence of either [142]. The large residual spatial fractions from our analyses also suggests the likely existence of additional geographically distributed factors related to overweight and obesity within in the Sydney Statistical Division.

The geographic variation observed in our data and reported in the substantive literature highlight the importance of appropriately controlling for spatial autocorrelation at analysis [39,42,61]. Spatial autocorrelation is problematic for standard regression methods that assume model residuals are independently and identically distributed (IDD), and its violation may result in biased inference [143]. Clustering is most typically handled by multilevel models that conflate unexplained spatial and non-spatial variation into a single random effect error term [42]. This approach addresses the issue of spatial autocorrelation; however, the potential value of the spatial variation for informing health planning is lost in the process. We have consistently identified variation in health risk-factors and outcomes in the Sydney region using 45 and Up Study data that indicate geographic areas with excess risk attributable to unobserved and spatially structured factors [39,42]. For example, we have reported variation in physical activity [42] and psychosocial distress [39] that indicates excess risk due to unobserved and spatially structured factors in addition to that attributable to observed individual-level factors, and postal area walkability and relative socioeconomic disadvantage. Pattenden and colleagues contend that outcome variation in excess of socioeconomic factors may indicate opportunities to address disparities in health status [61], while Fitzpatrick et al highlight its potential role in suggesting causal pathways [144]. We believe our approach is helpful because it not only locates inequalities in the geographic distribution of risk but also quantifies that attributable to known factors that may, or may not, be amenable to intervention, and to unknown factors requiring further investigation.

We observed statistically significant associations between most individual-level covariates and body mass outcomes in all fixed-effects models used to derive offset terms in spatial models. Consistent with our previous work on physical activity [42] and psychosocial distress [39] in this cohort, we observed strong positive associations between prevalence of high body mass and numbers of chronic care conditions ever diagnosed and recently treated, and even stronger associations with reduced functional capacity. These findings agree with previous reports on this cohort [103,120] and in the broader literature [145,146,147]. High body mass is considered a “gateway” into non-communicable diseases [148], and possibly multi-morbidity [149,150,151,152] and reduced physical functioning [145,153,154], although reverse causality is plausible with multi-morbidity and reduced physical functioning leading to lower levels of physical activity and poorer dietary choices [155]. We also observed an inverse association between high body mass and lower levels of psychosocial distress after adjusting for functional capacity. This is consistent with previous findings of increased risk of psychosocial distress with greater functional limitations in this cohort [39,156,157], and a strong contemporaneous effect of physical disability on depressive symptoms [158]. Depression and anxiety disorders are also known causes of weight loss in community-dwelling older adults [159], and may be influential on our findings as the Kessler Psychological Distress Scale [114] is specific for current anxiety and affective disorders in Australian community populations at the cut-points used in our study [160].

A major strength of our study is the large sample size drawn from the 45 and Up Study [93]. This high-quality, population-based cohort comprises approximately 10% of the Sydney statistical district aged 45 and over. While we make no claims to the external validity of our point-estimates beyond our sample, it is well established in the epidemiological literature that relative measures of effect are generalisable irrespective of representativeness and non response [161,162]. Methodological investigations of the 45 and Up Study cohort support this likelihood. Mealing et al. [130] have reported that odds ratios derived from the full cohort correspond to those from the population-representative New South Wales Adult Population Health Survey [163], while we have reported high correlations between postal area relative risks and disease maps estimated from unweighted and post-stratification weighted data [42,131]. These observations support the generalisability of our risk estimators and their geographical distribution to postal area populations within the Sydney Statistical Division area. We also used the Sydney Walkability Index as our exposure metric, which is derived using high-quality government agency spatial data [40]. The strengths of this index include its demonstrated predictive validity for moderate-intensity walking at multiple spatial scales, a cohesive latent variable structure, and comparability to other indexes (e.g., NQLS [38] and PLACE [19]) frequently used in walkability research [40,42]. The spatial data used in its construction are routinely updated to support NSW Government business, and are accessible via the *NSW Open Data Policy* [164]. This allows the Sydney Walkability Index to be re-calculated annually to monitor changes in the spatial distribution of walkability across the Sydney statistical district. There is also an ongoing effort to develop a national walkability index using similar methods to our index that would benchmark and monitor changes in walkability across Australia [28]. Finally, our study employed an explicitly spatial approach that controlled for individual-level factors to investigate geographic variation in high body mass and its association with environmental walkability at the postal area level. The substantial levels of clustering in our data indicate the importance of accounting for spatial autocorrelation in analyses where it is expected or observed, and highlights the potentially informative nature of this variation for health and urban planning that is ignored when spatial and non-spatial sources of variation are conflated [42].

Our study reported on associations between postal area walkability and high body mass outcomes, which are not necessarily causal. An important limitation of our study is that we were unable to control for potential bias due to participant self-selection into postal areas, which raises the potential for reverse causation. Self-selection bias occurs when individuals choose to live in neighbourhoods that support their physical activity and travel behaviour preferences [8,17]. Systematic reviews indicate that neighbourhood self-selection may account for up to 50% of the built environment’s effect on physical activity [17]. Its contribution to built environment associations with high body mass is less clear. Some studies have reported that self-selection fully accounts for these associations [165,166], while others have reported more modest attenuation effects [167,168,169]. There is also some evidence that it may selectively attenuate associations for continuous but not categorical body mass outcomes [170,171]. The 45 and Up Study does not collect information on respondents’ preferences for the neighbourhoods in which they reside, and so we are unable to discount this as contributing to some portion of the estimated effect of walkability in our study.

We used self-reported BMI to classify overweight and obesity, which is generally appropriate for quantifying body size in the 45 and Up Study cohort but known to underestimate prevalence of obesity using standard BMI classifications by 6% [98]. In the context of our study, this means both overweight and obesity are likely to have been systematically misclassified. Monte Carlo simulation studies have found that systematic misclassification of binary dependent variables on the order of 2–5% can bias relative effects estimates by 12–25% in either direction [172,173]. This has the potential to weaken the magnitude of our observed associations for both overweight and obesity, but would still result in meaningful effect sizes at the population-level [8,10]. Another limitation of our analysis is that it was conducted at a single geographic scale, and so our findings may differ if conducted at a finer or coarser scale. This is the Modifiable Areal Unit Problem [174,175], which is germane to all analyses using areal units or zones [175]. We were only provided with access to postal area identifiers by the data custodian, and so were unable to assess the sensitivity of our results to different spatial scales. We have previously examined associations between walkability and health-enhancing physical activity at different spatial scales and found similar relations [40,42], which provides some reassurance on the robustness of our findings to spatial scale. However, the influence of scale on matched exposure-response relations in the walkability literature remains opens and warrants further investigation.

The Sydney Walkability Index [40] is comparable to other indexes used in the substantive literature [19] and so is subject to the same limitation that walkability quartiles may be data-dependent as they are derived using population-specific cut-points [176,177]. We therefore encourage planners, policy makers and researchers to review the quartile cut-points used in constructing the Sydney Walkability Index to evaluate their appropriateness and the applicability of our findings to their geographic context. Further, modelling walkability as an index means we are unable to identify which built environment components in the index contribute to the observed associations with prevalence and geographic distribution of overweight and obesity, which would be useful for framing policy interventions. This was partly a choice for consistency with the expressed interest in the literature about the potential for “walkability indexes” to benchmark, inform and monitor development plans, but also because the added complexity would have made our models intractable. Our analysis used a two stage approach in which individual-level conditional probabilities of overweight and obesity were modelled first and then used as offset terms to adjust spatial models. While this approach is not uncommon in the epidemiological literature (see [39,42,122,123,124]), ideally we would have modelled individual and area-level effects simultaneously in a single, parsimonious model. However, despite the relative ease with which Besag, York and Mollié models can be fit in available software [75,178], they remain computationally prohibitive to implement outside of high performance computing environments when extended to multi-level problems comprising samples of the size used in our study [179]. Our units of analysis comprised Australian postal areas, which correspond in spatial extent to the upper limit of buffers sizes used in individual-level research linking walkability to high body mass but may not be representative of all spatial extents at which health and urban planning decisions occur. Finally, our study precludes causal inference due to its cross-sectional design.

## 5. Conclusions

Walkability indexes have been identified as potentially useful tools for planning and monitoring the built environment to improve health [27]. Our results provide support for their potential application to body mass outcomes by demonstrating that: (1) rates of overweight and obesity are negatively associated walkability at the postcode level for Sydney residents aged ≥45 years; and (2) that area-level walkability makes a small but meaningful contribution to the geographic clustering of high body mass in the Sydney metropolitan region. Our results also suggest the presence of other unobserved and spatially structured factors contributing to this clustering. *The Greater Sydney Region Plan* aims to create healthy, resilient and socially connected communities over the next 40 years by creating fine scaled urban form, mixed land use and amenity within walkable urban centres [25]. The methods and outcomes described here may assist in the geographical targeting of strategies and monitoring their progress towards achieving its liveability objectives.

## Figures and Tables

**Figure 1 ijerph-16-00664-f001:**
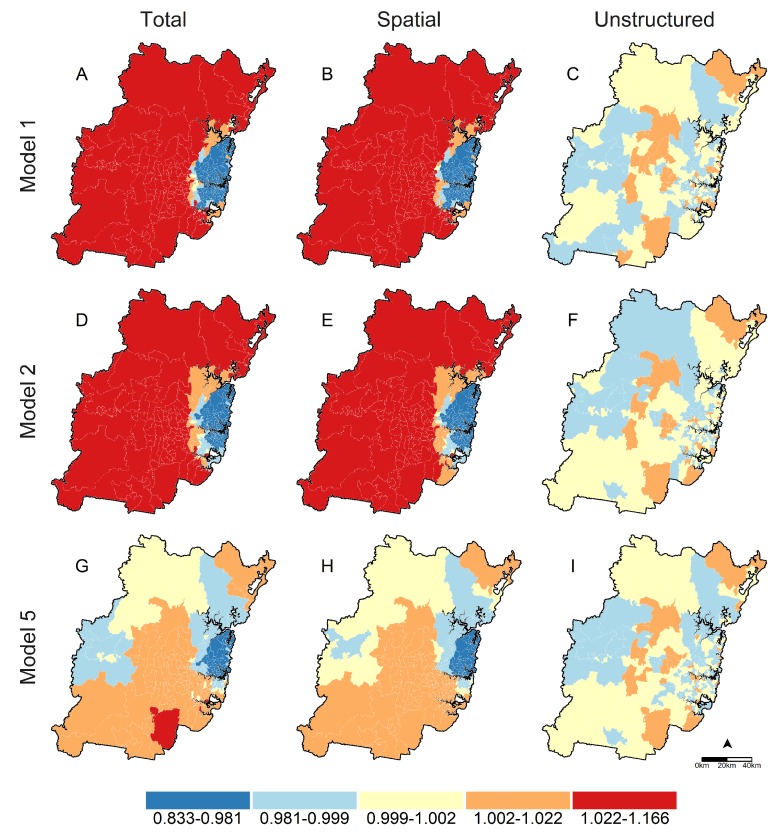
Total, Spatial and Unstructured prevalence ratios for overweight body mass in Sydney postal areas. Total prevalence ratios are derived by exponentiating the sum of spatial (*s*) and unstructured (*u*) random effects; Spatial and Unstructured prevalence ratios are obtained by exponentiating individual *s* and *u* components, respectively. Total, Spatial, and Unstructured prevalence ratio estimates are reported in maps (**A**–**C**) for Model 1, maps (**D**–**F**) for Model 2, and maps (**G**–**I**) for Model 5.

**Figure 2 ijerph-16-00664-f002:**
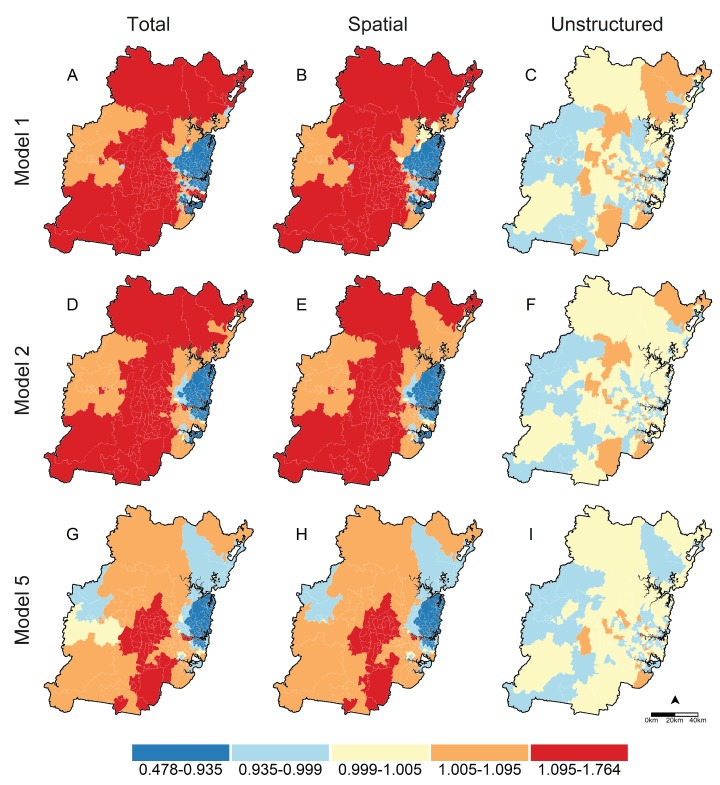
Total, Spatial and Unstructured prevalence ratios for obese body mass in Sydney postal areas. Total prevalence ratios are derived by exponentiating the sum of spatial (*s*) and unstructured (*u*) random effects; Spatial and Unstructured prevalence ratios are obtained by exponentiating individual *s* and *u* components, respectively. Total, Spatial, and Unstructured prevalence ratio estimates are reported in maps (**A**–**C**) for Model 1, maps (**D**–**F**) for Model 2, and maps (**G**–**I**) for Model 5.

**Table 1 ijerph-16-00664-t001:** Sample characteristics and prevalence of overweight and obesity among study participants.

Variable	Characteristics		Prevalence			
			Overweight		Obesity	
	N	%	n	%	n	%
AREA-LEVEL VARIABLES						
*Walkability*						
Low	25,454	27.6	10,150	52.9	6251	40.8
Low-medium	31,404	34.1	12,380	50.0	6655	35.0
Medium-high	19,449	21.1	7543	47.2	3454	29.0
High	15,850	17.2	5861	44.0	2516	25.2
*Socioeconomic disadvantage*						
Q1—Most	17,425	18.9	6697	52.1	4559	42.5
Q2	19,517	21.2	7579	51.7	4847	40.6
Q3—Middling	14,984	16.3	5877	49.4	3082	33.8
Q4	19,982	21.7	7938	47.8	3392	28.2
Q5—Least	20,249	22.0	7843	45.5	2996	24.1
INDIVIDUAL-LEVEL VARIABLES						
*Sex*						
Male	44,690	48.5	20,802	58.1	8912	37.3
Female	47,467	51.5	15,132	40.3	9964	30.8
*Age*						
45–49	13,550	14.7	4871	45.1	2761	31.8
50–54	16,723	18.1	6188	47.4	3665	34.8
55–59	16,717	18.1	6568	51.2	3885	38.3
60–64	13,742	14.9	5696	53.7	3136	39.0
65–69	10,188	11.1	4297	54.0	2227	37.8
70–74	6910	7.5	2969	53.3	1341	34.0
75–79	4999	5.4	2047	49.0	820	27.8
80–84	6614	7.2	2513	43.2	801	19.5
85+	2714	2.9	785	31.7	240	12.4
*Language spoken at home*						
English	78,028	84.7	30,768	49.9	16,330	34.6
Other	14,129	15.3	5166	44.6	2546	28.4
*Education level*						
Less than secondary school	7434	8.1	2704	50.6	2086	44.1
Secondary school graduation	26,741	29.0	10,171	49.2	6052	36.5
Trade, certificate or diploma	28,932	31.4	11,814	51.8	6143	35.9
University degree	29,050	31.5	11,245	46.0	4595	25.8
*Relationship status*						
Partner	68,759	74.6	27,826	50.7	13,863	33.9
No partner	23,398	25.4	8108	44.1	5013	32.8
*Employment status*						
Full-time work	32,716	35.5	13,622	53.5	7246	37.9
Part-time work	13,177	14.3	4418	41.0	2408	27.5
Other work	1358	1.5	426	39.6	281	30.2
Not working	44,906	48.7	17,468	48.6	8941	32.6
*Health insurance type*						
Private with extras	54,218	58.8	21,751	50.1	10,830	33.4
Private without extras	12,961	14.1	5058	47.2	2255	28.5
Government health care card	11,993	13.0	4351	47.8	2881	37.7
None	12,985	14.1	4774	47.4	2910	35.4
*Smoking status*						
Never smoked	54,117	58.7	20,518	46.6	10,072	30.0
Past smoker	31,639	34.3	13,145	54.2	7397	40.0
Current smoker	6401	6.9	2271	45.5	1407	34.1
*Psychosocial distress*						
Low	70,218	76.2	27,960	49.1	13,318	31.5
Moderate	14,573	15.8	5433	49.0	3475	38.0
High	5152	5.6	1828	48.4	1375	41.4
Very high	2214	2.4	713	47.3	708	47.2
*Diagnosed chronic conditions*						
0	31,297	34.0	11,955	44.1	4218	21.8
1	36,917	40.1	14,726	50.2	7560	34.1
2	18,186	19.7	7145	54.4	5040	45.6
3 or more	5757	6.2	2108	57.0	2058	56.4
*Treated chronic conditions*						
0	41,580	45.1	15,904	45.5	6590	25.7
1	30,121	32.7	12,141	51.3	6448	35.9
2	14,524	15.8	5721	53.5	3835	43.6
3 or more	5932	6.4	2168	55.2	2003	53.2
*Limited physical functioning*						
None	32,392	35.1	12,656	44.4	3908	19.8
Minor	25,125	27.3	10,628	52.4	4838	33.4
Moderate	20,316	22.0	7801	52.8	5555	44.4
Severe	14,324	15.5	4849	49.7	4575	48.3
SENSITIVITY VARIABLES						
*Total physical activity*						
0 min	5478	5.9	1868	50.9	1807	50.1
1–149 min	15,365	16.7	5895	52.1	4053	42.8
150–299 min	15,833	17.2	6241	50.5	3468	36.2
≥300 min	55,481	60.2	21,930	47.7	9548	28.5

N—Stratum total, n—Stratum outcome frequency, %—Stratum outcome per cent.

**Table 2 ijerph-16-00664-t002:** Adjusted odds ratios for individual-level analyses of overweight and obesity.

	Overweight		Obese	
	OR	95% CI	OR	95% CI
*Sex*	*p* < 0.0001		*p* < 0.0001	
Male	1.00		1.00	
Female	0.47	0.46–0.49	0.62	0.59–0.64
*Age*	*p* < 0.0001		*p* < 0.0001	
45–49	1.00		1.00	
50–54	1.00	0.95–1.05	0.94	0.88–1.00
55–59	1.07	1.01–1.13	0.90	0.84–0.97
60–64	1.08	1.02–1.15	0.76	0.70–0.82
65–69	1.00	0.93–1.07	0.59	0.54–0.65
70–74	0.87	0.81–0.94	0.39	0.35–0.43
75–79	0.66	0.60–0.72	0.23	0.21–0.26
80–84	0.50	0.46–0.54	0.12	0.11–0.14
85+	0.31	0.28–0.35	0.06	0.05–0.07
*Language spoken at home*	*p* < 0.0001		*p* < 0.0001	
English	1.00		1.00	
Other	0.81	0.78–0.84	0.72	0.68–0.77
*Education level*	*p* < 0.0001		*p* < 0.0001	
Less than secondary school	1.53	1.43–1.63	2.47	2.28–2.67
Secondary school graduation	1.35	1.29–1.40	1.77	1.67–1.86
Trade, certificate or diploma	1.27	1.22–1.32	1.54	1.46–1.62
University degree	1.00		1.00	
*Relationship status*	*p* < 0.0001		*p* = 0.1285	
Partner	1.00		1.00	
No partner	0.89	0.86–0.92	0.96	0.92–1.01
*Employment status*	*p* < 0.0001		*p* < 0.0001	
Full-time work	1.00		1.00	
Part-time work	0.75	0.71–0.79	0.61	0.57–0.65
Other work	0.72	0.64–0.82	0.61	0.52–0.71
Not working	0.78	0.75–0.82	0.66	0.62–0.70
*Health insurance type*	*p* < 0.0001		*p* < 0.0001	
Private with extras	1.00		1.00	
Private without extras	0.90	0.86–0.94	0.83	0.78–0.88
Government health care card	0.94	0.89–0.99	1.02	0.96–1.09
None	0.91	0.87–0.95	0.99	0.93–1.05
*Smoking status*	*p* < 0.0001		*p* < 0.0001	
Never smoked	1.00		1.00	
Past smoker	1.17	1.13–1.21	1.28	1.23–1.34
Current smoker	0.78	0.74–0.84	0.73	0.68–0.79
*Psychosocial distress*	*p* < 0.0001		*p* < 0.0001	
Low	1.00		1.00	
Moderate	0.94	0.90–0.98	0.91	0.86–0.96
High	0.88	0.82–0.95	0.82	0.76–0.89
Very high	0.83	0.74–0.92	0.88	0.78–1.00
*Diagnosed chronic conditions*	*p* < 0.0001		*p* < 0.0001	
0	1.00		1.00	
1	1.19	1.15–1.24	1.58	1.51–1.66
2	1.35	1.29–1.42	2.13	2.01–2.27
3 or more	1.48	1.37–1.60	2.69	2.46–2.93
*Treated chronic conditions*	*p* < 0.0001		*p* < 0.0001	
0	1.00		1.00	
1	1.22	1.18–1.27	1.47	1.40–1.54
2	1.38	1.31–1.45	1.89	1.77–2.01
3 or more	1.57	1.45–1.69	2.48	2.27–2.71
*Limited physical functioning*	*p* < 0.0001		*p* < 0.0001	
None	1.00		1.00	
Minor	1.36	1.30–1.41	2.10	1.99–2.21
Moderate	1.58	1.51–1.65	3.77	3.56–4.00
Severe	1.61	1.52–1.70	5.31	4.96–5.68

OR—Odds ratio, CI—Confidence interval.

**Table 3 ijerph-16-00664-t003:** Spatial regression summaries for postal area analyses of associations between overweight, walkability and relative socioeconomic disadvantage.

	Model 1	Model 2	Model 3	Model 4	Model 5
Individual-level adjustment	No	Yes	Yes	Yes	Yes
*Prevalence ratios (95% CrI)*					
Constant	0.99 (0.98–1.00)	1.00 (0.98–1.01)	1.03 (1.00–1.06)	1.01 (0.99–1.04)	1.07 (1.02–1.11)
Walkability					
Low	–	–	1.00	–	1.00
Low-medium	–	–	0.98 (0.95–1.01)	–	0.98 (0.95–1.01)
Medium-high	–	–	0.96 (0.92–1.00)	–	0.94 (0.91–0.98)
High	–	–	0.91 (0.87–0.97)	–	0.90 (0.86–0.94)
Socioeconomic disadvantage					
Q1—Most	–	–	–	1.00	1.00
Q2	–	–	–	1.01 (0.97–1.05)	1.01 (0.97–1.04)
Q3—Middling	–	–	–	0.99 (0.95–1.03)	0.99 (0.95–1.03)
Q4	–	–	–	0.97 (0.93–1.01)	0.97 (0.93–1.00)
Q5—Least	–	–	–	0.94 (0.90–0.99)	0.93 (0.89–0.97)
*Model diagnostics*					
pD	55.73	37.48	33.64	35.05	27.01
DIC	1832.77	1787.67	1787.12	1787.85	1782.70
Spatial fraction	0.965	0.932	0.882	0.900	0.673

CrI—credible interval, pD—effective parameters, DIC—Deviance Information Criterion. Model 1—null model with expected cases proportional to the overall prevalence. Model 2—null model with expected cases adjusted for individual-level factors. Model 3—Model 2 + Sydney Walkability Index. Model 4—Model 2 + Index of Relative Socioeconomic Disadvantage. Model 5—Model 3 + Index of Relative Socioeconomic Disadvantage.

**Table 4 ijerph-16-00664-t004:** Spatial regression summaries for postal area analyses of associations between obesity, walkability and relative socioeconomic disadvantage.

	Model 1	Model 2	Model 3	Model 4	Model 5
Individual-level adjustment	No	Yes	Yes	Yes	Yes
*Prevalence ratios (95% CrI)*					
Constant	0.95 (0.93–0.97)	0.96 (0.95–0.98)	1.02 (0.97–1.08)	1.01 (0.96–1.05)	1.10 (1.02–1.17)
Walkability					
Low	–	–	1.00	–	1.00
Low-medium	–	–	0.97 (0.91–1.02)	–	0.96 (0.91–1.01)
Medium-high	–	–	0.92 (0.85–0.99)	–	0.89 (0.83–0.96)
High	–	–	0.89 (0.80–0.99)	–	0.85 (0.78–0.94)
Socioeconomic disadvantage					
Q1—Most	–	–	–	1.00	1.00
Q2	–	–	–	1.03 (0.98–1.09)	1.02 (0.97–1.08)
Q3—Middling	–	–	–	0.97 (0.92–1.03)	0.97 (0.91–1.03)
Q4	–	–	–	0.91 (0.85–0.97)	0.90 (0.85–0.96)
Q5—Least	–	–	–	0.88 (0.82–0.95)	0.85 (0.79–0.92)
*Model diagnostics*					
pD	128.60	72.36	70.99	63.02	56.79
DIC	1794.88	1711.26	1712.90	1705.26	1703.00
Spatial fraction	0.992	0.985	0.981	0.978	0.961

CrI—credible interval, pD—effective parameters, DIC—Deviance Information Criterion. Model 1—null model with expected cases proportional to the overall prevalence. Model 1—null model with expected cases proportional to the overall prevalence. Model 3—Model 2 + Sydney Walkability Index. Model 4—Model 2 + Index of Relative Socioeconomic Disadvantage. Model 5—Model 3 + Index of Relative Socioeconomic Disadvantage.

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
