# Peer review of "Area-Level Walkability and the Geographic Distribution of High Body Mass in Sydney, Australia: A Spatial Analysis Using the 45 and Up Study"

_ijerph, 2019, doi:10.3390/ijerph16040664_

Round 1

Reviewer 1 Report

Review of ijerph-418406

The contribution of area-level walkability to geographic variation in high body mass in Sydney, Australia: a spatial analysis of 92,157 respondents to the 45 and Up Study

This is a useful and well-written paper. I have very few comments and support publication.

My first comment (question, really) is likely a result of my not being an epidemiologist, but rather someone in a social science field that has some important connections to the topic at hand… and therefore I am not accustomed to the methods used in epidemiology and request some clarification. I would appreciate some brief clarification on why the methods employed are (to my mind, coming from my field) unnecessarily aggregated to spatial units, when the principal actors involved in the causal chain we care about (walkability leads to walking leads to better health outcomes) are people, not spatial units. I suspect this is convention, or useful for making policy arguments, or something of that sort?

The question of residential self-selection should be better addressed. While I don’t suspect it’s a strong enough confounder to make the results *wrong*, it’s certainly plausible that a third thing (or set of things) is causing people to both locate in a walkable environment and also to be physically active. In my context (I suspect this may hold on some level in Australia), we are quite concerned with this confounding causal chain: many people wish to live in walkable environments, yet upon having a child, many move to less-walkable areas because of better schools, or because having a car and a place to park it is terrific when you have kids to cart around, etc, and thus perhaps part of the perceived effect of walkability on physical activity is simply the effect of having kids to cart around, having enough spare time to walk about for your everyday activities, etc. There are other potential confounders. This is somewhat mitigated by having a more homogenous sample in terms of life-cycle, but still there must be plausible alternative explanations for some portion of the estimated effect of walkability.

Please discuss, briefly, the bias introduced by self-reported weight in the BMI calculation. I believe there is compression at the top, right? Higher-BMI people are more apt to underreport weight?

Please explain why walkability is included as an index, rather than a set of variables. Wouldn’t it be useful to learn if street network connectivity, or, say, land use entropy is a stronger determinant of BMI? Depending on the answer, very different policy interventions may be effective.

Relatedly, some brief mention of the theory behind the three main elements of walkability within the index, as they relate to physical activity, would be useful. For instance, residential density is probably only useful in this paper as a proxy for other things – we simply do not visit other humans in their homes/apartments enough for the density of humans in their homes and apartments to make any real difference in physical activity. Proximity to a diversity of other locations has a much stronger theoretical connection to physical activity, since something like 70 percent of all travel is to places that are not (a) our own workplaces or (b) other peoples’ homes or apartments. Intersection density – the network distance approaches the straight-line distance as intersection density increases (i.e. it becomes more hypotenuse-y).

Further points on which I do not request any response:

The framing within the introduction is exceptionally good and succinct and quite a nice read.

While I do not really understand the rationale behind aggregating the findings, the spatial unit (Census postal areas) seems very reasonable.

It may be useful to give a bit more space to discussing the scale at which walkability has been measured in previous studies, as well as the theoretical underpinning of wanting a larger-scale, or multi-scalar walkability metric. Pulling from my living situation: my immediate surroundings are incredibly walkable, and I walk for groceries, lunch, and so forth, but absolutely everything else requires some sort of motorized transportation (in my case largely because of the barriers of exceptionally high crime beyond my immediate surroundings, an important part of walkability in some contexts). Many daily activities can reasonably be accomplished in walking distance, but (being in a very large region), many other activities (meeting friends, going to cultural events, the journey to work) require exiting the neighborhood, and thus some non-active travel mode.

The modeling appears appropriate from what I can tell. It’s a bit outside my modeling world so I’ll let others comment more on that.

Reviewer 2 Report

Overall, this is very well-written paper using robust methods to address an important topic related to both planning and public health. Its bibliography is impressive too. 

However, because the literature on the built environment's influence on BMI/overweight/obesity is extensive, this makes it important to clearly define where an article like this fits into the literature and how it fills important gaps or areas of uncertainty. This was an issue where I felt the paper needed some more work, as I think some of the areas they identified as gaps actually have a fair amount of coverage in the literature. Evaluating this was complicated a bit by the fact that  often the descriptions given by the authors of these research gaps were not very clear or precise, so it was hard to tell if some of the paper's claims of uniqueness were valid.  

As an example, on line 64 the authors write:  "Studies matching macro-level walkability exposures and body mass outcomes for populations are uncommon [16,17,40] despite calls from planners and policy makers for evidence at these spatial scales [39,41–43]."  Given how important this statement seems to be for the framing of the paper, there doesn't seem to be enough clarity or specificity to make the point convincingly. For instance, what is meant by "macro-scale" and why is it important to planners and policy makers to understand relationships at this scale? This should be explained, along with a bit more justification about why the extensive existing literature on walkability indices as related to BMI is insufficient to address the issues they raise. Doing so is critical to making the case for why this paper is making a needed contribution to the literature.

The paper also seems to be staking a claim to a unique niche in terms of spatial approach. However, this justification didn't seem convincing to me, especially where it's written starting line 83 : "However, few studies have addressed area-level geographic variation in overweight and obesity [44,47,50,52–62]; an even smaller number have considered built environment influences on this variation [50,54,55,62]; and only one appears to have evaluated walkability directly [54]." A similar claim comes up at the beginning of the Discussion.  

First of all, there are a large number of studies looking at such geographic variation, many of which are not cited here. See for instance, among many, many others:

Willms, J. Douglas, Mark S. Tremblay, and Peter T. Katzmarzyk. "Geographic and demographic variation in the prevalence of overweight Canadian children." Obesity Research 11.5 (2003): 668-673.

Sichieri, Rosely, et al. "High temporal, geographic, and income variation in body mass index among adults in Brazil." American Journal of Public Health 84.5 (1994): 793-798.

Zhuo, Qin, et al. "Geographic variation in the prevalence of overweight and economic status in Chinese adults." British journal of nutrition 102.3 (2009): 413-418.

Ackerson, Leland K., et al. "Geography of underweight and overweight among women in India: a multilevel analysis of 3204 neighborhoods in 26 states." Economics & Human Biology 6.2 (2008): 264-280.

Lebel, Alexandre, et al. "The geography of overweight in Quebec: a multilevel perspective." Canadian Journal of Public Health/Revue Canadienne de Sante'e Publique (2009): 18-23.

Second, the idea that walkability isn't covered in the literature on BMI/overweight  also seems inaccurate to me when we consider the large number of studies in this field, e.g. 

Brown, Barbara B., et al. "Mixed land use and walkability: Variations in land use measures and relationships with BMI, overweight, and obesity." Health & place 15.4 (2009): 1130-1141.

 Wasfi, Rania A., et al. "Neighborhood walkability and body mass index trajectories: longitudinal study of Canadians." American journal of public health 106.5 (2016): 934-940.

Smith, Ken R., et al. "Walkability and body mass index: density, design, and new diversity measures." American journal of preventive medicine 35.3 (2008): 237-244.

Smith, Ken R., et al. "Effects of neighborhood walkability on healthy weight: assessing selection and causal influences." Social science research 40.5 (2011): 1445-1455.

Wei, Yehua Dennis, et al. "Walkability, land use and physical activity." Sustainability 8.1 (2016): 65.

Maybe I'm missing something and the authors are referring to something more specific when they mention "area-level geographic variation," but I think a lot of the authors of the studies above would consider their work to deal with area-level geographic variation. And later, in the Discussion when it's written "Our study is novel because it directly addresses exposure-outcome relations at a geographic scale more proximal to those typically used by government agencies for population-level planning," the authors need to justify and clarify what they mean by this claim of uniqueness when many other authors might contest that. Doing so will require some more detailed discussion of the literature and this paper's relation to it. 

In other words, there is a fundamental problem in the the way the authors frame the paper in terms of its niche in the literature and the gaps/needs of that literature. Again, I may be misreading the intent of the authors, but if that's the case, it's because there is ambiguity in the way they frame this, and more detail is needed on what they mean by these supposed research gaps (e.g. what do they mean by "area level geographic variation"). Hence,  the authors either need to provide more convincingly articulated proof and verbal precision that what they propose is relatively unique, or they need to reframe the justification of the paper. To do such reframing means to modify the blanket statements about gaps in the research or literature such that there are no claims of insufficient research when that contention can't be supported. That may mean admitting that the paper is less unique than may have been contended, but that's OK--this paper still adds a valuable data point to the literature, and elements of it are unique in terms of setting and methods, and the authors should be clear about that. 

I believe that addressing the issues above in terms of contextualization within the literature will lead to a better articulation of the research objectives, which currently are a bit lacking in precision.  For instance, when they write that an objective is to "assess geographic variation in overweight and obesity of populations in Sydney at a spatial scale more typical of health planning," what exactly is meant by the latter part of that sentence, and how does that relate to needs and gaps in the literature? As written, the introduction does not shed light on that claim sufficiently so that the reader can see how such an objective relates to the literature. 

In terms of methods, the study seems robust and well executed. One small issue is that the authors should present standard deviations in addition to medians for postal code sizes and populations. This is important because it's critical to know if these spatial units are highly variable or consistent in size, population and characteristics.  If there is considerable variation, that leads to the potential confounding problem known as the "Modifiable Areal Unit Problem" (Openshaw 1984), which should be addressed in the paper, because it can lead to statistical bias and misleading results. 

Some other methodological issues that should be addressed:

-The authors mention that the spatial unit of analysis is a post code, but not how data were summarized at the post code level. I assume most individual level variables were averaged at the post code level, but this wasn't made explicit, which it should be. Also, how was walkability index summarized at the post code level? Is that the level that it is produced at, or was some kind of spatial analysis required to summarize at that level? If so, what is the minimum spatial unit of analysis at which the walkability index is calculated? This seems like an important methodological detail, since walkability the key predictor. 

-Why not conduct the analysis at the observational level of individual respondents geocoded by address, as other studies have done? Was this simply because of data limitations or is there a methodological reason to do so? 

-How consistent or clustered is the distribution of survey respondents by postal code? This relates back to the modifiable areal unit problem mentioned above.

-Why do a logistic regression where the dependent variable is binary (e.g. overweight or not), as opposed to doing a regression on a continuous BMI variable, as the latter would seem to yield more information (that is, to statistically distinguish between moderately and several overweight people)? The choice of models should be further explained.

-When the authors say "we fit six spatial regressions" is that referring to the BYM models or to some other type of spatial (e.g. spatial error or spatial lag) regression? The uncertainty relates to the fact that when the BYM model is presented, it is not referred to as a spatial regression. In either case this needs to be clarified. 

-The reason for including multiple statistical modeling approaches needs to be better discussed. For instance, why include a spatial regression approach? Is it meant to address bias introduced by spatial autocorrelation, or some other reason? This is discussed somewhat in the end of the Discussion, but really this justification needs to be in the Methods section, so readers understand why methods were chosen. Regardless, the statistical modeling section needs a bit more of a general introduction to summarize the general methods strategy before going into details. 

-Why not break up the walkability index into its components: building/intersection density and mix of uses, and using those as predictors, instead of modeling the actual index as the predictor? This should be justified. 

Finally, on the interpretation of results, I have two issues:

-I think the confounding and potentially interactive effects of socio-economic status could have been discussed more, particularly given that (if I'm reading correctly) some of the higher density areas of central Sydney  were found to have high clustering of overweight and obesity. Previous literature has found that the relationship between walkability and BMI is often contingent upon socio-economic status (SES), and that in some contexts, while BMI is negatively correlated with urban density/walkability in higher SES neighborhoods,   BMI can be positively correlated with with density in nearby low SES neighborhoods (see for instance,  Frank LD, Kerr J, Sallis JF, Miles R, Chapman J. A hierarchy of sociodemographic and environmental correlates of walking and obesity. Preventive Medicine. 2008;47(2):172–178. or  Rutt CD, Coleman KJ. Examining the relationships among built environment, physical activity, and body mass index in El Paso, TX. Preventive Medicine. 2005;40(6):831–841). I think that discussing this might help frame some of the interpretation of results and address what they refer to as "unobserved and spatially structured factors" such as in the case of central Sydney mentioned previously.  

-It should be made clear that association is not necessarily causation. As part of this discussion it should be mentioned that associations established through this research may be just as much due to self-selection of populations in a neighborhood as to the mechanism of walking. 

Reviewer 3 Report

This is a well-documented piece of research on the ecological association between walkability and body mass in Sydney, Australia. The paper is clear and well written.  I have nonetheless a few suggestions for improvement.

1. Rationale of the study

In the introduction, the authors suggest that there is a need for ecological research on the association between body mass and walkability at a spatial scale relevant to planners. I agree with the authors that it is important to conduct research based on tools used by planners so that it can be used to benchmark progress towards more active living neighbourhoods. In that sense, the use of the exposure variable is fully justified. However, the rationale for conducting an ecological study, whereas individual data are available, deserves better justification in my opinion. The authors indicate page 2 line 66 that there are calls from planners and policy makers for evidence at these spatial scales. This would need to be explained a bit more and better justified because population health is generally better assesses with individual-level data.

In addition, I would encourage the author to discuss further how changes in walkability are being monitored over time at the relevant geographical scale. This is important because the rationale of the study is to use walkability to monitor progress in planning strategies. A perquisite for this would be that regular data on walkability are collected.

2. Cross-sectional data and neighbourhood self-selection

Page 3 line 88-89, the authors indicate that the objective of the study is to “assess the potential of walkability indexes for planning and monitoring activities aimed at reducing population-levels of high body mass through built environment interventions”. I would encourage the authors to amend this statement given that the use of cross-sectional data prevents from establishing the directionality of the association between walkability and body mass. This means that the objective of the study cannot be met with the data at hand. An important issue in the field is the self-selection of individuals within neighbourhoods, based on neighbourhood characteristics. An association found between body mass and walkability can be bi-directional, and changes in walkability in some neighbourhood might lead to changes in neighbourhood compositions, which would have in turn little impact on population heath. This is briefly mentioned by the authors in the limitation section, but I believe that the interpretation of the results and the usability of monitoring walkability as a measure of progress towards improved population health should be discuss with more caution.   

3. Exposure variable

Could the author clearly indicate in relevant places when the data on walkability were collected. It is clearly stated that 2006 Census data are used. Is this also the year at which walkability data are calculated?

4. Statistical analyses

Although I am not familiar with all the statistical methods used, I find the statistical analysis section quite difficult to follow, and I would strongly encourage the authors to redraft it in order to increase the clarity. In particular:

-         It would help if the authors would provide model equations for the models used to calculate predicted probabilities.

-         What do the authors mean by “mean and covariate adjusted fixed-effect logistic regression models”? Are the authors using conditional fixed effects models?

-         I find it difficult to understand which covariates are included in each model.

2. Generalisability

The first sentence of the abstract states that “improving the walkability of the built environment…. is increasingly considered in regional development plans”. However, a lot of the literature cited in the introduction (eg. P2 lines 44-47) comes from Australia. I am therefore wondering whether the authors could provide evidence from other settings or refine their statements in their introduction/abstract to mention that their study is of particular interest to the Australian context.

In addition, I am wondering what could be the generalisability of the results, given that inconsistencies of associations between walkability and body mass around the globe, and the specificity of the index used, which seems to be developed for Sydney. Would planner use different tools to measure walkability in different parts of Australia? Are they also predictive of body mass? Are tools used in other settings comparable? This is something that would deserved to be discussed.

Other minor comments:

Page 4, line 136: the study cited (ref 73), found that BMI calculated with self-reported data is under-estimated. The authors might mention this in addition to saying that the measure was validated.

Page 4, line  137: I would not use ‘study variable’ but rather ‘exposure variable’, which is much more commonly used in the field (and that the author use the term exposure in other places).

Round 2

Reviewer 1 Report

Thank you for addressing my concerns. I continue to believe that individual-level analysis would be more appropriate, but I see the merits outlined in your response to my first round of comments, and defer to the other referees and to the editors in this matter.

Reviewer 2 Report

Good job addressing suggested revisions.